# ALBERT: A Lite BERT for Self-supervised Learning of Language Representations

**Zhenzhong Lan**[1]     **Mingda Chen**[2*]     **Sebastian Goodman**[1]     **Kevin Gimpel**[2]

**Piyush Sharma**[1]     **Radu Soricut**[1]

[1]Google Research        [2]Toyota Technological Institute at Chicago

`{lanzhzh, seabass, piyushsharma, rsoricut}@google.com`
`{mchen, kgimpel}@ttic.edu`

## ABSTRACT

Increasing model size when pretraining natural language representations often results in improved performance on downstream tasks. However, at some point further model increases become harder due to GPU/TPU memory limitations and longer training times. To address these problems, we present two parameter-reduction techniques to lower memory consumption and increase the training speed of BERT (Devlin et al., 2019). Comprehensive empirical evidence shows that our proposed methods lead to models that scale much better compared to the original BERT. We also use a self-supervised loss that focuses on modeling inter-sentence coherence, and show it consistently helps downstream tasks with multi-sentence inputs. As a result, our best model establishes new state-of-the-art results on the GLUE, RACE, and SQuAD benchmarks while having fewer parameters compared to BERT-large. The code and the pretrained models are available at `https://github.com/google-research/ALBERT`.

## 1 INTRODUCTION

Full network pre-training (Dai & Le, 2015; Radford et al., 2018; Devlin et al., 2019; Howard & Ruder, 2018) has led to a series of breakthroughs in language representation learning. Many non-trivial NLP tasks, including those that have limited training data, have greatly benefited from these pre-trained models. One of the most compelling signs of these breakthroughs is the evolution of machine performance on a reading comprehension task designed for middle and high-school English exams in China, the RACE test (Lai et al., 2017): the paper that originally describes the task and formulates the modeling challenge reports then state-of-the-art machine accuracy at 44.1%; the latest published result reports their model performance at 83.2% (Liu et al., 2019); the work we present here pushes it even higher to 89.4%, a stunning 45.3% improvement that is mainly attributable to our current ability to build high-performance pretrained language representations.

Evidence from these improvements reveals that a large network is of crucial importance for achieving state-of-the-art performance (Devlin et al., 2019; Radford et al., 2019). It has become common practice to pre-train large models and distill them down to smaller ones (Sun et al., 2019; Turc et al., 2019) for real applications. Given the importance of model size, we ask: *Is having better NLP models as easy as having larger models*?

An obstacle to answering this question is the memory limitations of available hardware. Given that current state-of-the-art models often have hundreds of millions or even billions of parameters, it is easy to hit these limitations as we try to scale our models. Training speed can also be significantly hampered in distributed training, as the communication overhead is directly proportional to the number of parameters in the model.

Existing solutions to the aforementioned problems include model parallelization (Shazeer et al., 2018; Shoeybi et al., 2019) and clever memory management (Chen et al., 2016; Gomez et al., 2017).

---

*Work done as an intern at Google Research, driving data processing and downstream task evaluations.

These solutions address the memory limitation problem, but not the communication overhead. In this paper, we address all of the aforementioned problems, by designing A Lite BERT (ALBERT) architecture that has significantly fewer parameters than a traditional BERT architecture.

ALBERT incorporates two parameter reduction techniques that lift the major obstacles in scaling pre-trained models. The first one is a factorized embedding parameterization. By decomposing the large vocabulary embedding matrix into two small matrices, we separate the size of the hidden layers from the size of vocabulary embedding. This separation makes it easier to grow the hidden size without significantly increasing the parameter size of the vocabulary embeddings. The second technique is cross-layer parameter sharing. This technique prevents the parameter from growing with the depth of the network. Both techniques significantly reduce the number of parameters for BERT without seriously hurting performance, thus improving parameter-efficiency. An ALBERT configuration similar to BERT-large has 18x fewer parameters and can be trained about 1.7x faster. The parameter reduction techniques also act as a form of regularization that stabilizes the training and helps with generalization.

To further improve the performance of ALBERT, we also introduce a self-supervised loss for sentence-order prediction (SOP). SOP primary focuses on inter-sentence coherence and is designed to address the ineffectiveness (Yang et al., 2019; Liu et al., 2019) of the next sentence prediction (NSP) loss proposed in the original BERT.

As a result of these design decisions, we are able to scale up to much larger ALBERT configurations that still have fewer parameters than BERT-large but achieve significantly better performance. We establish new state-of-the-art results on the well-known GLUE, SQuAD, and RACE benchmarks for natural language understanding. Specifically, we push the RACE accuracy to $89.4\%$, the GLUE benchmark to 89.4, and the F1 score of SQuAD 2.0 to 92.2.

## 2 RELATED WORK

### 2.1 SCALING UP REPRESENTATION LEARNING FOR NATURAL LANGUAGE

Learning representations of natural language has been shown to be useful for a wide range of NLP tasks and has been widely adopted (Mikolov et al., 2013; Le & Mikolov, 2014; Dai & Le, 2015; Peters et al., 2018; Devlin et al., 2019; Radford et al., 2018; 2019). One of the most significant changes in the last two years is the shift from pre-training word embeddings, whether standard (Mikolov et al., 2013; Pennington et al., 2014) or contextualized (McCann et al., 2017; Peters et al., 2018), to full-network pre-training followed by task-specific fine-tuning (Dai & Le, 2015; Radford et al., 2018; Devlin et al., 2019). In this line of work, it is often shown that larger model size improves performance. For example, Devlin et al. (2019) show that across three selected natural language understanding tasks, using larger hidden size, more hidden layers, and more attention heads always leads to better performance. However, they stop at a hidden size of 1024, presumably because of the model size and computation cost problems.

It is difficult to experiment with large models due to computational constraints, especially in terms of GPU/TPU memory limitations. Given that current state-of-the-art models often have hundreds of millions or even billions of parameters, we can easily hit memory limits. To address this issue, Chen et al. (2016) propose a method called gradient checkpointing to reduce the memory requirement to be sublinear at the cost of an extra forward pass. Gomez et al. (2017) propose a way to reconstruct each layer's activations from the next layer so that they do not need to store the intermediate activations. Both methods reduce the memory consumption at the cost of speed. Raffel et al. (2019) proposed to use model parallelization to train a giant model. In contrast, our parameter-reduction techniques reduce memory consumption and increase training speed.

### 2.2 CROSS-LAYER PARAMETER SHARING

The idea of sharing parameters across layers has been previously explored with the Transformer architecture (Vaswani et al., 2017), but this prior work has focused on training for standard encoder-decoder tasks rather than the pretraining/finetuning setting. Different from our observations, Dehghani et al. (2018) show that networks with cross-layer parameter sharing (Universal Transformer, UT) get better performance on language modeling and subject-verb agreement than the standard

transformer. Very recently, Bai et al. (2019) propose a Deep Equilibrium Model (DQE) for transformer networks and show that DQE can reach an equilibrium point for which the input embedding and the output embedding of a certain layer stay the same. Our observations show that our embeddings are oscillating rather than converging. Hao et al. (2019) combine a parameter-sharing transformer with the standard one, which further increases the number of parameters of the standard transformer.

## 2.3 SENTENCE ORDERING OBJECTIVES

ALBERT uses a pretraining loss based on predicting the ordering of two consecutive segments of text. Several researchers have experimented with pretraining objectives that similarly relate to discourse coherence. Coherence and cohesion in discourse have been widely studied and many phenomena have been identified that connect neighboring text segments (Hobbs, 1979; Halliday & Hasan, 1976; Grosz et al., 1995). Most objectives found effective in practice are quite simple. Skip-thought (Kiros et al., 2015) and FastSent (Hill et al., 2016) sentence embeddings are learned by using an encoding of a sentence to predict words in neighboring sentences. Other objectives for sentence embedding learning include predicting future sentences rather than only neighbors (Gan et al., 2017) and predicting explicit discourse markers (Jernite et al., 2017; Nie et al., 2019). Our loss is most similar to the sentence ordering objective of Jernite et al. (2017), where sentence embeddings are learned in order to determine the ordering of two consecutive sentences. Unlike most of the above work, however, our loss is defined on textual segments rather than sentences. BERT (Devlin et al., 2019) uses a loss based on predicting whether the second segment in a pair has been swapped with a segment from another document. We compare to this loss in our experiments and find that sentence ordering is a more challenging pretraining task and more useful for certain downstream tasks. Concurrently to our work, Wang et al. (2019) also try to predict the order of two consecutive segments of text, but they combine it with the original next sentence prediction in a three-way classification task rather than empirically comparing the two.

## 3 THE ELEMENTS OF ALBERT

In this section, we present the design decisions for ALBERT and provide quantified comparisons against corresponding configurations of the original BERT architecture (Devlin et al., 2019).

## 3.1 MODEL ARCHITECTURE CHOICES

The backbone of the ALBERT architecture is similar to BERT in that it uses a transformer encoder (Vaswani et al., 2017) with GELU nonlinearities (Hendrycks & Gimpel, 2016). We follow the BERT notation conventions and denote the vocabulary embedding size as $E$, the number of encoder layers as $L$, and the hidden size as $H$. Following Devlin et al. (2019), we set the feed-forward/filter size to be $4H$ and the number of attention heads to be $H/64$.

There are three main contributions that ALBERT makes over the design choices of BERT.

**Factorized embedding parameterization.** In BERT, as well as subsequent modeling improvements such as XLNet (Yang et al., 2019) and RoBERTa (Liu et al., 2019), the WordPiece embedding size $E$ is tied with the hidden layer size $H$, i.e., $E \equiv H$. This decision appears suboptimal for both modeling and practical reasons, as follows.

From a modeling perspective, WordPiece embeddings are meant to learn *context-independent* representations, whereas hidden-layer embeddings are meant to learn *context-dependent* representations. As experiments with context length indicate (Liu et al., 2019), the power of BERT-like representations comes from the use of context to provide the signal for learning such context-dependent representations. As such, untying the WordPiece embedding size $E$ from the hidden layer size $H$ allows us to make a more efficient usage of the total model parameters as informed by modeling needs, which dictate that $H \gg E$.

From a practical perspective, natural language processing usually require the vocabulary size $V$ to be large.[1] If $E \equiv H$, then increasing $H$ increases the size of the embedding matrix, which has size

---

[1]Similar to BERT, all the experiments in this paper use a vocabulary size $V$ of 30,000.

$V \times E$. This can easily result in a model with billions of parameters, most of which are only updated sparsely during training.

Therefore, for ALBERT we use a factorization of the embedding parameters, decomposing them into two smaller matrices. Instead of projecting the one-hot vectors directly into the hidden space of size $H$, we first project them into a lower dimensional embedding space of size $E$, and then project it to the hidden space. By using this decomposition, we reduce the embedding parameters from $O(V \times H)$ to $O(V \times E + E \times H)$. This parameter reduction is significant when $H \gg E$. We choose to use the same E for all word pieces because they are much more evenly distributed across documents compared to whole-word embedding, where having different embedding size (Grave et al. (2017); Baevski & Auli (2018); Dai et al. (2019) ) for different words is important.

**Cross-layer parameter sharing.** For ALBERT, we propose cross-layer parameter sharing as another way to improve parameter efficiency. There are multiple ways to share parameters, e.g., only sharing feed-forward network (FFN) parameters across layers, or only sharing attention parameters. The default decision for ALBERT is to share all parameters across layers. All our experiments use this default decision unless otherwise specified. We compare this design decision against other strategies in our experiments in Sec. 4.5.

Similar strategies have been explored by Dehghani et al. (2018) (Universal Transformer, UT) and Bai et al. (2019) (Deep Equilibrium Models, DQE) for Transformer networks. Different from our observations, Dehghani et al. (2018) show that UT outperforms a vanilla Transformer. Bai et al. (2019) show that their DQEs reach an equilibrium point for which the input and output embedding of a certain layer stay the same. Our measurement on the L2 distances and cosine similarity show that our embeddings are oscillating rather than converging.

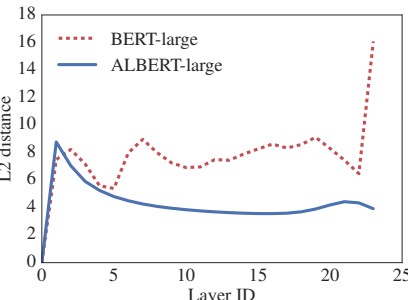 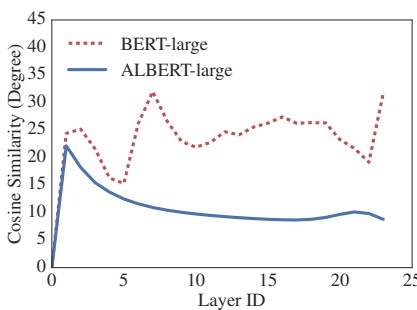

Figure 1: The L2 distances and cosine similarity (in terms of degree) of the input and output embedding of each layer for BERT-large and ALBERT-large.

Figure 1 shows the L2 distances and cosine similarity of the input and output embeddings for each layer, using BERT-large and ALBERT-large configurations (see Table 1). We observe that the transitions from layer to layer are much smoother for ALBERT than for BERT. These results show that weight-sharing has an effect on stabilizing network parameters. Although there is a drop for both metrics compared to BERT, they nevertheless do not converge to 0 even after 24 layers. This shows that the solution space for ALBERT parameters is very different from the one found by DQE.

**Inter-sentence coherence loss.** In addition to the masked language modeling (MLM) loss (Devlin et al., 2019), BERT uses an additional loss called next-sentence prediction (NSP). NSP is a binary classification loss for predicting whether two segments appear consecutively in the original text, as follows: positive examples are created by taking consecutive segments from the training corpus; negative examples are created by pairing segments from different documents; positive and negative examples are sampled with equal probability. The NSP objective was designed to improve performance on downstream tasks, such as natural language inference, that require reasoning about the relationship between sentence pairs. However, subsequent studies (Yang et al., 2019; Liu et al., 2019) found NSP's impact unreliable and decided to eliminate it, a decision supported by an improvement in downstream task performance across several tasks.

We conjecture that the main reason behind NSP's ineffectiveness is its lack of difficulty as a task, as compared to MLM. As formulated, NSP conflates *topic prediction* and *coherence prediction* in a

| Model | | Parameters | Layers | Hidden | Embedding | Parameter-sharing |
|---|---|---|---|---|---|---|
| BERT | base | 108M | 12 | 768 | 768 | False |
| | large | 334M | 24 | 1024 | 1024 | False |
| ALBERT | base | 12M | 12 | 768 | 128 | True |
| | large | 18M | 24 | 1024 | 128 | True |
| | xlarge | 60M | 24 | 2048 | 128 | True |
| | xxlarge | 235M | 12 | 4096 | 128 | True |

Table 1: The configurations of the main BERT and ALBERT models analyzed in this paper.

single task[2]. However, topic prediction is easier to learn compared to coherence prediction, and also overlaps more with what is learned using the MLM loss.

We maintain that inter-sentence modeling is an important aspect of language understanding, but we propose a loss based primarily on *coherence*. That is, for ALBERT, we use a sentence-order prediction (SOP) loss, which avoids topic prediction and instead focuses on modeling inter-sentence coherence. The SOP loss uses as positive examples the same technique as BERT (two consecutive segments from the same document), and as negative examples the same two consecutive segments but with their order swapped. This forces the model to learn finer-grained distinctions about discourse-level coherence properties. As we show in Sec. 4.6, it turns out that NSP cannot solve the SOP task at all (i.e., it ends up learning the easier topic-prediction signal, and performs at random-baseline level on the SOP task), while SOP can solve the NSP task to a reasonable degree, presumably based on analyzing misaligned coherence cues. As a result, ALBERT models consistently improve downstream task performance for multi-sentence encoding tasks.

## 3.2 MODEL SETUP

We present the differences between BERT and ALBERT models with comparable hyperparameter settings in Table 1. Due to the design choices discussed above, ALBERT models have much smaller parameter size compared to corresponding BERT models.

For example, ALBERT-large has about 18x fewer parameters compared to BERT-large, 18M versus 334M. An ALBERT-xlarge configuration with $H = 2048$ has only 60M parameters and an ALBERT-xxlarge configuration with $H = 4096$ has 233M parameters, i.e., around 70% of BERT-large's parameters. Note that for ALBERT-xxlarge, we mainly report results on a 12-layer network because a 24-layer network (with the same configuration) obtains similar results but is computationally more expensive.

This improvement in parameter efficiency is the most important advantage of ALBERT's design choices. Before we can quantify this advantage, we need to introduce our experimental setup in more detail.

## 4 EXPERIMENTAL RESULTS

## 4.1 EXPERIMENTAL SETUP

To keep the comparison as meaningful as possible, we follow the BERT (Devlin et al., 2019) setup in using the BOOKCORPUS (Zhu et al., 2015) and English Wikipedia (Devlin et al., 2019) for pretraining baseline models. These two corpora consist of around 16GB of uncompressed text. We format our inputs as "[CLS] $x_1$ [SEP] $x_2$ [SEP]", where $x_1 = x_{1,1}, x_{1,2} \cdots$ and $x_2 = x_{1,1}, x_{1,2} \cdots$ are two segments.[3] We always limit the maximum input length to 512, and randomly generate input sequences shorter than 512 with a probability of 10%. Like BERT, we use a vocabulary size of 30,000, tokenized using SentencePiece (Kudo & Richardson, 2018) as in XLNet (Yang et al., 2019).

---

[2]Since a negative example is constructed using material from a different document, the negative-example segment is misaligned both from a topic and from a coherence perspective.

[3]A segment is usually comprised of more than one natural sentence, which has been shown to benefit performance by Liu et al. (2019).

We generate masked inputs for the MLM targets using $n$-gram masking (Joshi et al., 2019), with the length of each $n$-gram mask selected randomly. The probability for the length $n$ is given by

$$p(n) = \frac{1/n}{\sum_{k=1}^{N} 1/k}$$

We set the maximum length of $n$-gram (i.e., $n$) to be 3 (i.e., the MLM target can consist of up to a 3-gram of complete words, such as "White House correspondents").

All the model updates use a batch size of 4096 and a LAMB optimizer with learning rate 0.00176 (You et al., 2019). We train all models for 125,000 steps unless otherwise specified. Training was done on Cloud TPU V3. The number of TPUs used for training ranged from 64 to 512, depending on model size.

The experimental setup described in this section is used for all of our own versions of BERT as well as ALBERT models, unless otherwise specified.

## 4.2 EVALUATION BENCHMARKS

### 4.2.1 INTRINSIC EVALUATION

To monitor the training progress, we create a development set based on the development sets from SQuAD and RACE using the same procedure as in Sec. 4.1. We report accuracies for both MLM and sentence classification tasks. Note that we only use this set to check how the model is converging; it has not been used in a way that would affect the performance of any downstream evaluation, such as via model selection.

### 4.2.2 DOWNSTREAM EVALUATION

Following Yang et al. (2019) and Liu et al. (2019), we evaluate our models on three popular benchmarks: The General Language Understanding Evaluation (GLUE) benchmark (Wang et al., 2018), two versions of the Stanford Question Answering Dataset (SQuAD; Rajpurkar et al., 2016; 2018), and the ReAding Comprehension from Examinations (RACE) dataset (Lai et al., 2017). For completeness, we provide description of these benchmarks in Appendix A.3. As in (Liu et al., 2019), we perform early stopping on the development sets, on which we report all comparisons except for our final comparisons based on the task leaderboards, for which we also report test set results. For GLUE datasets that have large variances on the dev set, we report median over 5 runs.

## 4.3 OVERALL COMPARISON BETWEEN BERT AND ALBERT

We are now ready to quantify the impact of the design choices described in Sec. 3, specifically the ones around parameter efficiency. The improvement in parameter efficiency showcases the most important advantage of ALBERT's design choices, as shown in Table 2: with only around 70% of BERT-large's parameters, ALBERT-xxlarge achieves significant improvements over BERT-large, as measured by the difference on development set scores for several representative downstream tasks: SQuAD v1.1 (+1.9%), SQuAD v2.0 (+3.1%), MNLI (+1.4%), SST-2 (+2.2%), and RACE (+8.4%).

Another interesting observation is the speed of data throughput at training time under the same training configuration (same number of TPUs). Because of less communication and fewer computations, ALBERT models have higher data throughput compared to their corresponding BERT models. If we use BERT-large as the baseline, we observe that ALBERT-large is about 1.7 times faster in iterating through the data while ALBERT-xxlarge is about 3 times slower because of the larger structure.

Next, we perform ablation experiments that quantify the individual contribution of each of the design choices for ALBERT.

## 4.4 FACTORIZED EMBEDDING PARAMETERIZATION

Table 3 shows the effect of changing the vocabulary embedding size $E$ using an ALBERT-base configuration setting (see Table 1), using the same set of representative downstream tasks. Under the non-shared condition (BERT-style), larger embedding sizes give better performance, but not by

| Model | | Parameters | SQuAD1.1 | SQuAD2.0 | MNLI | SST-2 | RACE | Avg | Speedup |
|---|---|---|---|---|---|---|---|---|---|
| BERT | base | 108M | 90.4/83.2 | 80.4/77.6 | 84.5 | 92.8 | 68.2 | 82.3 | 4.7x |
| | large | 334M | 92.2/85.5 | 85.0/82.2 | 86.6 | 93.0 | 73.9 | 85.2 | 1.0 |
| ALBERT | base | 12M | 89.3/82.3 | 80.0/77.1 | 81.6 | 90.3 | 64.0 | 80.1 | 5.6x |
| | large | 18M | 90.6/83.9 | 82.3/79.4 | 83.5 | 91.7 | 68.5 | 82.4 | 1.7x |
| | xlarge | 60M | 92.5/86.1 | 86.1/83.1 | 86.4 | 92.4 | 74.8 | 85.5 | 0.6x |
| | xxlarge | 235M | **94.1/88.3** | **88.1/85.1** | **88.0** | **95.2** | 82.3 | **88.7** | 0.3x |

Table 2: Dev set results for models pretrained over BOOKCORPUS and Wikipedia for 125k steps. Here and everywhere else, the Avg column is computed by averaging the scores of the downstream tasks to its left (the two numbers of F1 and EM for each SQuAD are first averaged).

much. Under the all-shared condition (ALBERT-style), an embedding of size 128 appears to be the best. Based on these results, we use an embedding size $E = 128$ in all future settings, as a necessary step to do further scaling.

| Model | $E$ | Parameters | SQuAD1.1 | SQuAD2.0 | MNLI | SST-2 | RACE | Avg |
|---|---|---|---|---|---|---|---|---|
| ALBERT base not-shared | 64 | 87M | 89.9/82.9 | 80.1/77.8 | 82.9 | 91.5 | 66.7 | 81.3 |
| | 128 | 89M | 89.9/82.8 | 80.3/77.3 | 83.7 | 91.5 | 67.9 | 81.7 |
| | 256 | 93M | 90.2/83.2 | 80.3/77.4 | 84.1 | 91.9 | 67.3 | 81.8 |
| | 768 | 108M | 90.4/83.2 | 80.4/77.6 | 84.5 | 92.8 | 68.2 | 82.3 |
| ALBERT base all-shared | 64 | 10M | 88.7/81.4 | 77.5/74.8 | 80.8 | 89.4 | 63.5 | 79.0 |
| | 128 | 12M | 89.3/82.3 | 80.0/77.1 | 81.6 | 90.3 | 64.0 | 80.1 |
| | 256 | 16M | 88.8/81.5 | 79.1/76.3 | 81.5 | 90.3 | 63.4 | 79.6 |
| | 768 | 31M | 88.6/81.5 | 79.2/76.6 | 82.0 | 90.6 | 63.3 | 79.8 |

Table 3: The effect of vocabulary embedding size on the performance of ALBERT-base.

## 4.5 CROSS-LAYER PARAMETER SHARING

Table 4 presents experiments for various cross-layer parameter-sharing strategies, using an ALBERT-base configuration (Table 1) with two embedding sizes ($E = 768$ and $E = 128$). We compare the all-shared strategy (ALBERT-style), the not-shared strategy (BERT-style), and intermediate strategies in which only the attention parameters are shared (but not the FNN ones) or only the FFN parameters are shared (but not the attention ones).

The all-shared strategy hurts performance under both conditions, but it is less severe for $E = 128$ (-1.5 on Avg) compared to $E = 768$ (-2.5 on Avg). In addition, most of the performance drop appears to come from sharing the FFN-layer parameters, while sharing the attention parameters results in no drop when $E = 128$ (+0.1 on Avg), and a slight drop when $E = 768$ (-0.7 on Avg).

There are other strategies of sharing the parameters cross layers. For example, We can divide the $L$ layers into $N$ groups of size $M$, and each size-$M$ group shares parameters. Overall, our experimental results shows that the smaller the group size $M$ is, the better the performance we get. However, decreasing group size $M$ also dramatically increase the number of overall parameters. We choose all-shared strategy as our default choice.

| Model | | Parameters | SQuAD1.1 | SQuAD2.0 | MNLI | SST-2 | RACE | Avg |
|---|---|---|---|---|---|---|---|---|
| ALBERT base $E$=768 | all-shared | 31M | 88.6/81.5 | 79.2/76.6 | 82.0 | 90.6 | 63.3 | 79.8 |
| | shared-attention | 83M | 89.9/82.7 | 80.0/77.2 | 84.0 | 91.4 | 67.7 | 81.6 |
| | shared-FFN | 57M | 89.2/82.1 | 78.2/75.4 | 81.5 | 90.8 | 62.6 | 79.5 |
| | not-shared | 108M | 90.4/83.2 | 80.4/77.6 | 84.5 | 92.8 | 68.2 | 82.3 |
| ALBERT base $E$=128 | all-shared | 12M | 89.3/82.3 | 80.0/77.1 | 82.0 | 90.3 | 64.0 | 80.1 |
| | shared-attention | 64M | 89.9/82.8 | 80.7/77.9 | 83.4 | 91.9 | 67.6 | 81.7 |
| | shared-FFN | 38M | 88.9/81.6 | 78.6/75.6 | 82.3 | 91.7 | 64.4 | 80.2 |
| | not-shared | 89M | 89.9/82.8 | 80.3/77.3 | 83.2 | 91.5 | 67.9 | 81.6 |

Table 4: The effect of cross-layer parameter-sharing strategies, ALBERT-base configuration.

## 4.6 SENTENCE ORDER PREDICTION (SOP)

We compare head-to-head three experimental conditions for the additional inter-sentence loss: none (XLNet- and RoBERTa-style), NSP (BERT-style), and SOP (ALBERT-style), using an ALBERT-base configuration. Results are shown in Table 5, both over intrinsic (accuracy for the MLM, NSP, and SOP tasks) and downstream tasks.

| | Intrinsic Tasks | | | Downstream Tasks | | | | | |
|---|---|---|---|---|---|---|---|---|---|
| SP tasks | MLM | NSP | SOP | SQuAD1.1 | SQuAD2.0 | MNLI | SST-2 | RACE | Avg |
| None | 54.9 | 52.4 | 53.3 | 88.6/81.5 | 78.1/75.3 | 81.5 | 89.9 | 61.7 | 79.0 |
| NSP | 54.5 | 90.5 | 52.0 | 88.4/81.5 | 77.2/74.6 | 81.6 | **91.1** | 62.3 | 79.2 |
| SOP | 54.0 | 78.9 | 86.5 | **89.3/82.3** | **80.0/77.1** | **82.0** | 90.3 | **64.0** | **80.1** |

Table 5: The effect of sentence-prediction loss, NSP vs. SOP, on intrinsic and downstream tasks.

The results on the intrinsic tasks reveal that the NSP loss brings no discriminative power to the SOP task (52.0% accuracy, similar to the random-guess performance for the "None" condition). This allows us to conclude that NSP ends up modeling only topic shift. In contrast, the SOP loss does solve the NSP task relatively well (78.9% accuracy), and the SOP task even better (86.5% accuracy). Even more importantly, the SOP loss appears to consistently improve downstream task performance for multi-sentence encoding tasks (around +1% for SQuAD1.1, +2% for SQuAD2.0, +1.7% for RACE), for an Avg score improvement of around +1%.

## 4.7 WHAT IF WE TRAIN FOR THE SAME AMOUNT OF TIME?

The speed-up results in Table 2 indicate that data-throughput for BERT-large is about 3.17x higher compared to ALBERT-xxlarge. Since longer training usually leads to better performance, we perform a comparison in which, instead of controlling for data throughput (number of training steps), we control for the actual training time (i.e., let the models train for the same number of hours). In Table 6, we compare the performance of a BERT-large model after 400k training steps (after 34h of training), roughly equivalent with the amount of time needed to train an ALBERT-xxlarge model with 125k training steps (32h of training).

| Models | Steps | Time | SQuAD1.1 | SQuAD2.0 | MNLI | SST-2 | RACE | Avg |
|---|---|---|---|---|---|---|---|---|
| BERT-large | 400k | 34h | 93.5/87.4 | 86.9/84.3 | 87.8 | 94.6 | 77.3 | 87.2 |
| ALBERT-xxlarge | 125k | 32h | **94.0/88.1** | **88.3/85.3** | 87.8 | **95.4** | **82.5** | **88.7** |

Table 6: The effect of controlling for training time, BERT-large vs ALBERT-xxlarge configurations.

After training for roughly the same amount of time, ALBERT-xxlarge is significantly better than BERT-large: +1.5% better on Avg, with the difference on RACE as high as +5.2%.

## 4.8 ADDITIONAL TRAINING DATA AND DROPOUT EFFECTS

The experiments done up to this point use only the Wikipedia and BOOKCORPUS datasets, as in (Devlin et al., 2019). In this section, we report measurements on the impact of the additional data used by both XLNet (Yang et al., 2019) and RoBERTa (Liu et al., 2019).

Fig. 2a plots the dev set MLM accuracy under two conditions, without and with additional data, with the latter condition giving a significant boost. We also observe performance improvements on the downstream tasks in Table 7, except for the SQuAD benchmarks (which are Wikipedia-based, and therefore are negatively affected by out-of-domain training material).

| | SQuAD1.1 | SQuAD2.0 | MNLI | SST-2 | RACE | Avg |
|---|---|---|---|---|---|---|
| No additional data | **89.3/82.3** | **80.0/77.1** | 81.6 | 90.3 | 64.0 | 80.1 |
| With additional data | 88.8/81.7 | 79.1/76.3 | **82.4** | **92.8** | **66.0** | **80.8** |

Table 7: The effect of additional training data using the ALBERT-base configuration.

We also note that, even after training for 1M steps, our largest models still do not overfit to their training data. As a result, we decide to remove dropout to further increase our model capacity. The

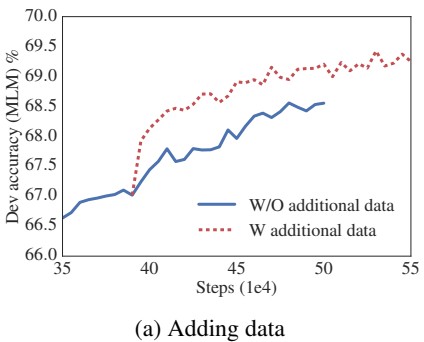

(a) Adding data

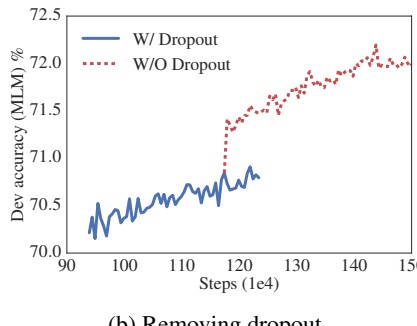

(b) Removing dropout

Figure 2: The effects of adding data and removing dropout during training.

plot in Fig. 2b shows that removing dropout significantly improves MLM accuracy. Intermediate evaluation on ALBERT-xxlarge at around 1M training steps (Table 8) also confirms that removing dropout helps the downstream tasks. There is empirical (Szegedy et al., 2017) and theoretical (Li et al., 2019) evidence showing that a combination of batch normalization and dropout in Convolutional Neural Networks may have harmful results. To the best of our knowledge, we are the first to show that dropout can hurt performance in large Transformer-based models. However, the underlying network structure of ALBERT is a special case of the transformer and further experimentation is needed to see if this phenomenon appears with other transformer-based architectures or not.

|  | SQuAD1.1 | SQuAD2.0 | MNLI | SST-2 | RACE | Avg |
|---|---|---|---|---|---|---|
| With dropout | 94.7/89.2 | 89.6/86.9 | 90.0 | 96.3 | 85.7 | 90.4 |
| Without dropout | **94.8/89.5** | **89.9/87.2** | **90.4** | **96.5** | **86.1** | **90.7** |

Table 8: The effect of removing dropout, measured for an ALBERT-xxlarge configuration.

## 4.9 CURRENT STATE-OF-THE-ART ON NLU TASKS

The results we report in this section make use of the training data used by Devlin et al. (2019), as well as the additional data used by Liu et al. (2019) and Yang et al. (2019). We report state-of-the-art results under two settings for fine-tuning: single-model and ensembles. In both settings, we only do single-task fine-tuning[4]. Following Liu et al. (2019), on the development set we report the median result over five runs.

| Models | MNLI | QNLI | QQP | RTE | SST | MRPC | CoLA | STS | WNLI | Avg |
|---|---|---|---|---|---|---|---|---|---|---|
| *Single-task single models on dev* | | | | | | | | | | |
| BERT-large | 86.6 | 92.3 | 91.3 | 70.4 | 93.2 | 88.0 | 60.6 | 90.0 | - | - |
| XLNet-large | 89.8 | 93.9 | 91.8 | 83.8 | 95.6 | 89.2 | 63.6 | 91.8 | - | - |
| RoBERTa-large | 90.2 | 94.7 | **92.2** | 86.6 | 96.4 | **90.9** | 68.0 | 92.4 | - | - |
| ALBERT (1M) | 90.4 | 95.2 | 92.0 | 88.1 | 96.8 | 90.2 | 68.7 | 92.7 | - | - |
| ALBERT (1.5M) | **90.8** | **95.3** | **92.2** | **89.2** | **96.9** | **90.9** | **71.4** | **93.0** | - | - |
| *Ensembles on test (from leaderboard as of Sept. 16, 2019)* | | | | | | | | | | |
| ALICE | 88.2 | 95.7 | **90.7** | 83.5 | 95.2 | 92.6 | **69.2** | 91.1 | 80.8 | 87.0 |
| MT-DNN | 87.9 | 96.0 | 89.9 | 86.3 | 96.5 | 92.7 | 68.4 | 91.1 | 89.0 | 87.6 |
| XLNet | 90.2 | 98.6 | 90.3 | 86.3 | 96.8 | 93.0 | 67.8 | 91.6 | 90.4 | 88.4 |
| RoBERTa | 90.8 | 98.9 | 90.2 | 88.2 | 96.7 | 92.3 | 67.8 | 92.2 | 89.0 | 88.5 |
| Adv-RoBERTa | 91.1 | 98.8 | 90.3 | 88.7 | 96.8 | 93.1 | 68.0 | 92.4 | 89.0 | 88.8 |
| ALBERT | **91.3** | **99.2** | 90.5 | **89.2** | **97.1** | **93.4** | 69.1 | **92.5** | **91.8** | **89.4** |

Table 9: State-of-the-art results on the GLUE benchmark. For single-task single-model results, we report ALBERT at 1M steps (comparable to RoBERTa) and at 1.5M steps. The ALBERT ensemble uses models trained with 1M, 1.5M, and other numbers of steps.

The single-model ALBERT configuration incorporates the best-performing settings discussed: an ALBERT-xxlarge configuration (Table 1) using combined MLM and SOP losses, and no dropout.

---

[4]Following Liu et al. (2019), we fine-tune for RTE, STS, and MRPC using an MNLI checkpoint.

The checkpoints that contribute to the final ensemble model are selected based on development set performance; the number of checkpoints considered for this selection range from 6 to 17, depending on the task. For the GLUE (Table 9) and RACE (Table 10) benchmarks, we average the model predictions for the ensemble models, where the candidates are fine-tuned from different training steps using the 12-layer and 24-layer architectures. For SQuAD (Table 10), we average the prediction scores for those spans that have multiple probabilities; we also average the scores of the "unanswerable" decision.

Both single-model and ensemble results indicate that ALBERT improves the state-of-the-art significantly for all three benchmarks, achieving a GLUE score of 89.4, a SQuAD 2.0 test F1 score of 92.2, and a RACE test accuracy of 89.4. The latter appears to be a particularly strong improvement, a jump of +17.4% absolute points over BERT (Devlin et al., 2019; Clark et al., 2019), +7.6% over XLNet (Yang et al., 2019), +6.2% over RoBERTa (Liu et al., 2019), and 5.3% over DCMI+ (Zhang et al., 2019), an ensemble of multiple models specifically designed for reading comprehension tasks. Our single model achieves an accuracy of $86.5\%$, which is still $2.4\%$ better than the state-of-the-art ensemble model.

| Models | SQuAD1.1 dev | SQuAD2.0 dev | SQuAD2.0 test | RACE test (Middle/High) |
|---|---|---|---|---|
| *Single model (from leaderboard as of Sept. 23, 2019)* | | | | |
| BERT-large | 90.9/84.1 | 81.8/79.0 | 89.1/86.3 | 72.0 (76.6/70.1) |
| XLNet | 94.5/89.0 | 88.8/86.1 | 89.1/86.3 | 81.8 (85.5/80.2) |
| RoBERTa | 94.6/88.9 | 89.4/86.5 | 89.8/86.8 | 83.2 (86.5/81.3) |
| UPM | - | - | 89.9/87.2 | - |
| XLNet + SG-Net Verifier++ | - | - | 90.1/87.2 | - |
| ALBERT (1M) | 94.8/89.2 | 89.9/87.2 | - | 86.0 (88.2/85.1) |
| ALBERT (1.5M) | **94.8/89.3** | **90.2/87.4** | **90.9/88.1** | **86.5 (89.0/85.5)** |
| *Ensembles (from leaderboard as of Sept. 23, 2019)* | | | | |
| BERT-large | 92.2/86.2 | - | - | - |
| XLNet + SG-Net Verifier | - | - | 90.7/88.2 | - |
| UPM | - | - | 90.7/88.2 | |
| XLNet + DAAF + Verifier | - | - | 90.9/88.6 | |
| DCMN+ | - | - | - | 84.1 (88.5/82.3) |
| ALBERT | **95.5/90.1** | **91.4/88.9** | **92.2/89.7** | **89.4 (91.2/88.6)** |

Table 10: State-of-the-art results on the SQuAD and RACE benchmarks.

# 5 DISCUSSION

While ALBERT-xxlarge has less parameters than BERT-large and gets significantly better results, it is computationally more expensive due to its larger structure. An important next step is thus to speed up the training and inference speed of ALBERT through methods like sparse attention (Child et al., 2019) and block attention (Shen et al., 2018). An orthogonal line of research, which could provide additional representation power, includes hard example mining (Mikolov et al., 2013) and more efficient language modeling training (Yang et al., 2019). Additionally, although we have convincing evidence that sentence order prediction is a more consistently-useful learning task that leads to better language representations, we hypothesize that there could be more dimensions not yet captured by the current self-supervised training losses that could create additional representation power for the resulting representations.

ACKNOWLEDGEMENT

The authors would like to thank Beer Changpinyo, Nan Ding, Noam Shazeer, and Tomer Levinboim for discussion and providing useful feedback on the project; Omer Levy and Naman Goyal for clarifying experimental setup for RoBERTa; Zihang Dai for clarifying XLNet; Brandon Norick, Emma Strubell, Shaojie Bai, Chas Leichner, and Sachin Mehta for providing useful feedback on the paper; Jacob Devlin for providing the English and multilingual version of training data; Liang Xu, Chenjie Cao and the CLUE community for providing the training data and evaluation benechmark of the Chinese version of ALBERT models.

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

# A  APPENDIX

## A.1  EFFECT OF NETWORK DEPTH AND WIDTH

In this section, we check how depth (number of layers) and width (hidden size) affect the performance of ALBERT. Table 11 shows the performance of an ALBERT-large configuration (see Table 1) using different numbers of layers. Networks with 3 or more layers are trained by fine-tuning using the parameters from the depth before (e.g., the 12-layer network parameters are fine-tuned from the checkpoint of the 6-layer network parameters).[5] Similar technique has been used in Gong et al. (2019). If we compare a 3-layer ALBERT model with a 1-layer ALBERT model, although they have the same number of parameters, the performance increases significantly. However, there are diminishing returns when continuing to increase the number of layers: the results of a 12-layer network are relatively close to the results of a 24-layer network, and the performance of a 48-layer network appears to decline.

| Number of layers | Parameters | SQuAD1.1 | SQuAD2.0 | MNLI | SST-2 | RACE | Avg |
|---|---|---|---|---|---|---|---|
| 1 | 18M | 31.1/22.9 | 50.1/50.1 | 66.4 | 80.8 | 40.1 | 52.9 |
| 3 | 18M | 79.8/69.7 | 64.4/61.7 | 77.7 | 86.7 | 54.0 | 71.2 |
| 6 | 18M | 86.4/78.4 | 73.8/71.1 | 81.2 | 88.9 | 60.9 | 77.2 |
| 12 | 18M | 89.8/83.3 | 80.7/77.9 | 83.3 | 91.7 | 66.7 | 81.5 |
| 24 | 18M | **90.3/83.3** | **81.8/79.0** | 83.3 | 91.5 | **68.7** | **82.1** |
| 48 | 18M | 90.0/83.1 | **81.8/78.9** | **83.4** | **91.9** | 66.9 | 81.8 |

Table 11: The effect of increasing the number of layers for an ALBERT-large configuration.

A similar phenomenon, this time for width, can be seen in Table 12 for a 3-layer ALBERT-large configuration. As we increase the hidden size, we get an increase in performance with diminishing returns. At a hidden size of 6144, the performance appears to decline significantly. We note that none of these models appear to overfit the training data, and they all have higher training and development loss compared to the best-performing ALBERT configurations.

---

[5]If we compare the performance of ALBERT-large here to the performance in Table 2, we can see that this warm-start technique does not help to improve the downstream performance. However, it does help the 48-layer network to converge. A similar technique has been applied to our ALBERT-xxlarge, where we warm-start from a 6-layer network.

| Hidden size | Parameters | SQuAD1.1 | SQuAD2.0 | MNLI | SST-2 | RACE | Avg |
|---|---|---|---|---|---|---|---|
| 1024 | 18M | 79.8/69.7 | 64.4/61.7 | 77.7 | 86.7 | 54.0 | 71.2 |
| 2048 | 60M | 83.3/74.1 | 69.1/66.6 | 79.7 | 88.6 | 58.2 | 74.6 |
| 4096 | 225M | **85.0/76.4** | **71.0/68.1** | **80.3** | **90.4** | **60.4** | **76.3** |
| 6144 | 499M | 84.7/75.8 | 67.8/65.4 | 78.1 | 89.1 | 56.0 | 74.0 |

Table 12: The effect of increasing the hidden-layer size for an ALBERT-large 3-layer configuration.

## A.2 Do very wide ALBERT models need to be deep(er) too?

In Section A.1, we show that for ALBERT-large ($H$=1024), the difference between a 12-layer and a 24-layer configuration is small. Does this result still hold for much wider ALBERT configurations, such as ALBERT-xxlarge ($H$=4096)?

| Number of layers | SQuAD1.1 | SQuAD2.0 | MNLI | SST-2 | RACE | Avg |
|---|---|---|---|---|---|---|
| 12 | 94.0/88.1 | 88.3/85.3 | 87.8 | 95.4 | 82.5 | 88.7 |
| 24 | 94.1/88.3 | 88.1/85.1 | 88.0 | 95.2 | 82.3 | 88.7 |

Table 13: The effect of a deeper network using an ALBERT-xxlarge configuration.

The answer is given by the results from Table 13. The difference between 12-layer and 24-layer ALBERT-xxlarge configurations in terms of downstream accuracy is negligible, with the Avg score being the same. We conclude that, when sharing all cross-layer parameters (ALBERT-style), there is no need for models deeper than a 12-layer configuration.

## A.3 Downstream Evaluation Tasks

**GLUE** GLUE is comprised of 9 tasks, namely Corpus of Linguistic Acceptability (CoLA; Warstadt et al., 2018), Stanford Sentiment Treebank (SST; Socher et al., 2013), Microsoft Research Paraphrase Corpus (MRPC; Dolan & Brockett, 2005), Semantic Textual Similarity Benchmark (STS; Cer et al., 2017), Quora Question Pairs (QQP; Iyer et al., 2017), Multi-Genre NLI (MNLI; Williams et al., 2018), Question NLI (QNLI; Rajpurkar et al., 2016), Recognizing Textual Entailment (RTE; Dagan et al., 2005; Bar-Haim et al., 2006; Giampiccolo et al., 2007; Bentivogli et al., 2009) and Winograd NLI (WNLI; Levesque et al., 2012). It focuses on evaluating model capabilities for natural language understanding. When reporting MNLI results, we only report the "match" condition (MNLI-m). We follow the finetuning procedures from prior work (Devlin et al., 2019; Liu et al., 2019; Yang et al., 2019) and report the held-out test set performance obtained from GLUE submissions. For test set submissions, we perform task-specific modifications for WNLI and QNLI as described by Liu et al. (2019) and Yang et al. (2019).

**SQuAD** SQuAD is an extractive question answering dataset built from Wikipedia. The answers are segments from the context paragraphs and the task is to predict answer spans. We evaluate our models on two versions of SQuAD: v1.1 and v2.0. SQuAD v1.1 has 100,000 human-annotated question/answer pairs. SQuAD v2.0 additionally introduced 50,000 unanswerable questions. For SQuAD v1.1, we use the same training procedure as BERT, whereas for SQuAD v2.0, models are jointly trained with a span extraction loss and an additional classifier for predicting answerability (Yang et al., 2019; Liu et al., 2019). We report both development set and test set performance.

**RACE** RACE is a large-scale dataset for multi-choice reading comprehension, collected from English examinations in China with nearly 100,000 questions. Each instance in RACE has 4 candidate answers. Following prior work (Yang et al., 2019; Liu et al., 2019), we use the concatenation of the passage, question, and each candidate answer as the input to models. Then, we use the representations from the "[CLS]" token for predicting the probability of each answer. The dataset consists of two domains: middle school and high school. We train our models on both domains and report accuracies on both the development set and test set.

## A.4 HYPERPARAMETERS

Hyperparameters for downstream tasks are shown in Table 14. We adapt these hyperparameters from Liu et al. (2019), Devlin et al. (2019), and Yang et al. (2019).

|  | LR | BSZ | ALBERT DR | Classifier DR | TS | WS | MSL |
|---|---|---|---|---|---|---|---|
| CoLA | 1.00E-05 | 16 | 0 | 0.1 | 5336 | 320 | 512 |
| STS | 2.00E-05 | 16 | 0 | 0.1 | 3598 | 214 | 512 |
| SST-2 | 1.00E-05 | 32 | 0 | 0.1 | 20935 | 1256 | 512 |
| MNLI | 3.00E-05 | 128 | 0 | 0.1 | 10000 | 1000 | 512 |
| QNLI | 1.00E-05 | 32 | 0 | 0.1 | 33112 | 1986 | 512 |
| QQP | 5.00E-05 | 128 | 0.1 | 0.1 | 14000 | 1000 | 512 |
| RTE | 3.00E-05 | 32 | 0.1 | 0.1 | 800 | 200 | 512 |
| MRPC | 2.00E-05 | 32 | 0 | 0.1 | 800 | 200 | 512 |
| WNLI | 2.00E-05 | 16 | 0.1 | 0.1 | 2000 | 250 | 512 |
| SQuAD v1.1 | 5.00E-05 | 48 | 0 | 0.1 | 3649 | 365 | 384 |
| SQuAD v2.0 | 3.00E-05 | 48 | 0 | 0.1 | 8144 | 814 | 512 |
| RACE | 2.00E-05 | 32 | 0.1 | 0.1 | 12000 | 1000 | 512 |

Table 14: Hyperparameters for ALBERT in downstream tasks. LR: Learning Rate. BSZ: Batch Size. DR: Dropout Rate. TS: Training Steps. WS: Warmup Steps. MSL: Maximum Sequence Length.

