# OpenReview forum: "ALBERT: A Lite BERT for Self-supervised Learning of Language Representations"
_ICLR.cc/2020/Conference — Accept (Spotlight)_

### Official Review · AnonReviewer3 · 2019-10-21
**Official Blind Review #3**

**Rating:** 6

**Review:**

The authors present ALBERT, a modification of the BERT architecture with substantially fewer parameters. They show that despite being much smaller, the performance is very strong and achieves state of the art on a variety of different tasks. There are several ideas proposed here: embedding factorization, sharing layers, and sentence ordering as a training objective.

1. The point that naively increasing the size of the BERT architecture does not work is a good one, but the authors don't acknowledge that this is tied up in the effect of regularization. Cross layer parameter sharing has a regularization effect that simply scaling up BERT large to x-large or such sizes does not have. This is also an issue with the authors making the statement that they are the first to show that dropout is harmful for Transformers. This is a large generalization that seems to be a special case of not only the regularized architecture they propose but also the large quantity of data that the model still underfits to.

2. The authors propose embedding factorization to reduce the number of parameters in the embedding dimension. This is very intuitive, but the authors do not cite or compare to related approaches. I understand these models are computationally intensive and thus do not expect large quantities of detailed ablations. However, this kind of dimensionality reduction has been explored with other techniques, for example for knowledge distillation, quantization, or even adaptive input/softmax (and with subword as well, not just whole word modeling). These techniques have also been applied to machine translation models, which do not use them to learn rare words. I believe a better discussion of these methods should be added to the paper, as this is not a novel proposition.

3. A large takeaway I have from this paper is that parameter size is not a good metric. While ALBERT is substantially smaller, the authors do not make it clear that this model is very slow at inference time due to the large size. This raises several questions: is it better to have models that are deeper or more wide? Can the authors actually report the latency in a comparative table next to BERT? Can the authors provide a sense of how large this model is in MB - e.g. presumably a goal of less parameters would be to have a model with less memory, but then the decision between memory and latency that different models make should be made more clear.

4. Section 4.8 is not clear. Exactly how much data, in terms of GB of uncompressed text, is used here? Is it the data of XLNet and RoBERTa, so larger than both of those settings individually? Further, the authors train for 1 million steps. This is larger than both XLNet and RoBERTa, is that correct? Or there is some detail about the size of the batch that actually makes it comparable? The many small tables where the changes are not clearly delineated makes it difficult to compare results.


**Experience Assessment:**

I have published in this field for several years.

**Review Assessment: Checking Correctness Of Derivations And Theory:**

I assessed the sensibility of the derivations and theory.

**Review Assessment: Checking Correctness Of Experiments:**

I assessed the sensibility of the experiments.

**Review Assessment: Thoroughness In Paper Reading:**

I read the paper thoroughly.

---

> ### Author Response · Authors · 2019-11-07
> **Thank you so much for going through the paper carefully and providing positive and useful feedback to our work**
>
> Dear Reviewer #3,
>
> Thank you so much for going through the paper carefully and providing positive and useful feedback to our work!
> Please see our responses below:
>
> We do mention that “The parameter reduction techniques also act as a form of regularization that stabilizes the training and helps with generalization.” However, we did not want to interpret along this dimension too much, as we did not have rigorous proof that regularization can solve the model degradation problem. One of the major difficulties in doing experiments with the BERT-xlarge setup is that it needs to train on 1024 TPUs v3 units, which is an expensive resource.
>
> For the overgeneralization problem in the sentence “dropout can hurt performance in large Transformer-based models”, we will make it clear by adding this sentence: “However, the underlying network structure of ALBERT is a special case of the Transformer, and further experimentation is needed to see if this phenomenon appears with other Transformer-based variants.”
>
> We agree that a better discussion of related work would help people to better understand our work.  Currently, we have a public comment from Sachin Mehta asking to compare related works such as ‘Adaptive input representations for neural language modeling.’, ‘transformer-xl’, and ‘Efficient softmax approximation for GPUs’. Please let us know if you have any specific additional work that you want us to include in the comparison of our embedding factorization methods, and we will incorporate it in the final version of the paper.
>
> The reason we did not report the inference time (latency) is because it is a platform-specific metric. We would need different strategies to optimize for TPUs and CPUs. However, we do have some TPU-based metrics for BERT-base and ALBERT-base. By looking at these numbers, we see that ALBERT-base is about 3x faster than BERT-base at inference time. Other people have also converted a Chinese version of ALBERT-tiny (we would like to thank brightmart for implementing this project and he got an amazing number (1.4k) of stars in such a short amount of time!) into tf-lite format and measured the inference time on mobile devices. Here is the quote from his website (https://github.com/brightmart/albert_zh): “On an Android phone w/ Qualcomm's SD845 SoC, via the above benchmark tool, as of 2019/11/01, the inference latency is ~120ms w/ this converted TFLite model using 4 threads on CPU, and the memory usage is ~60MB for the model during inference. Note the performance will improve further with future TFLite implementation optimizations.”
>
> That being said, we would say ALBERT could have a huge impact on inference speed because inference speed is usually memory-bandwidth bound and memory bandwidth is limited by memory capacity. For example, if you can keep the model weights used in matmuls in a smaller high-bandwidth memory, you can go 10x - 100x faster than if you need to read it out of a larger lower-bandwidth memory.
>
> In terms of model size in MB, they are roughly 4x as large as the parameter size as the weights are mostly in float format.
>
> We use all the XLNet data (126G) as well as the stories data (31G) of raw data. Our 1M step training went though the same number of iterations over the data as Roberta 500K, as they use 2x as large a batch size as ours.
>
> Thanks for bringing up these points, they are certainly valid and relevant. We will incorporate more info along the discussion above in the next version of our paper.

---

### Official Review · AnonReviewer1 · 2019-10-22
**Official Blind Review #1**

**Rating:** 8

**Review:**

Summary: This paper investigates improving upon BERT by reducing complexity in terms of free parameters and memory footprint as well as computation steps. They propose 2 strategies for doing this: 1) Splitting the embedding matrix into two smaller matrices (going from V x A to V x B + B x A where B <<<< A); 2) layer-wise parameter sharing. They also utilize sentence order prediction to help with training. These coupled with a bunch of other choices such as using the lamb optimizer, certain hyperparameters etc help show dramatic empirical gains across the board on a wide variety of NLP/NLU tasks.

Positives: This paper has a dramatic, seemingly statistically significant reduction in error across a wide-variety of tasks. It provides a thorough experimental plan and approaches the few addendums to training (splitting the embedding matrix, the layer-wise parameter sharing, and the sentence order prediction).

Concerns & Questions: There's a lot of experimentation here and a lot of seemingly deliberate choices after seeing empirical results during the research phase. How crucial are the choices of optimizer and other specific hyperparameters? Were there ones you observed that were more brittle than others? Any specific 'reasonable' configurations/settings that caused degenerate solutions?

**Experience Assessment:**

I have published one or two papers in this area.

**Review Assessment: Checking Correctness Of Derivations And Theory:**

I assessed the sensibility of the derivations and theory.

**Review Assessment: Checking Correctness Of Experiments:**

I assessed the sensibility of the experiments.

**Review Assessment: Thoroughness In Paper Reading:**

I read the paper at least twice and used my best judgement in assessing the paper.

---

> ### Author Response · Authors · 2019-11-07
> **Thank you so much for going through the paper carefully and providing such a positive feedback about our work**
>
> Dear Reviewer # 1
>
> Thank you so much for going through the paper carefully and providing such a positive feedback about our work! Please see our answers below:
>
> For hyper-parameter tuning, we only explore those hyper-parameters that are related to model size. This is done for the following two reasons: 1) To keep the comparison as meaningful as possible (so we fixed all other parameters as in BERT, and always used LAMB optimizer) 2) To keep under control the number of experiments we need to run; we already have a lot of experiments to report on; if we were to tune other hyper-parameters like learning rate and optimizer, the number of experiments needed can easily run out of control.
> For optimizer, we choose LAMB because it allows us to use large batch sizes. We haven’t tested other optimizers yet.
> Because large models that can cause degenerate solutions are extremely expensive to run, we did not explore that area very much. For example, in order to run with a batch size of 4096, BERT-xlarge requires 1024 TPUs v3 units, which is an expensive resource. However, we are working on this and hopefully can give a reasonable explanation/solution to this problem soon.

---

> > ### Comment · AnonReviewer1 · 2019-11-15
> > **Reply to authors**
> >
> > Okay thanks for the clarification!

---

### Official Review · AnonReviewer2 · 2019-10-25
**Official Blind Review #2**

**Rating:** 8

**Review:**

This paper proposes a new pre-trained BERT-like model called ALBERT. The contributions are mainly 3-fold: factorized embedding parameterization, cross-layer parameter sharing, and intern-sentence coherence loss. The first two address the issue of model size and memory consumption in BERT; the third corresponds to a new auxiliary task in pre-train, sentence-order prediction (SOP), replacing the next sentence prediction (NSP) task in BERT. These modifications lead to a much leaner model and improved performance. As a result, ALBERT pushes the state of the art on GLUE, RACE, and SQuAD while having fewer parameters than BERT-large.

This is a well-written paper which is easy to follow even for readers without deep background knowledge. The proposed method is meaningful and effective. Its empirical results are impressive.

Other comments:

- Section 4.9. Why use the all-share condition for state-of-the-art ALBERT results (as indicated in Table 2)? Judging from Table 4 and 5, shouldn't the non-shared condition give better results? The number of parameters would be larger, of course.

- I like the justification/motivation given for replacing NSP with SOP. I wonder if the authors have tried other objectives (but didn't work out). Such negative results are valuable to practitioners.

- Typo in Sec. 4.1: x1,1, x1,2 should be x2,1, x2,2.


**Experience Assessment:**

I have published one or two papers in this area.

**Review Assessment: Checking Correctness Of Derivations And Theory:**

N/A

**Review Assessment: Checking Correctness Of Experiments:**

I assessed the sensibility of the experiments.

**Review Assessment: Thoroughness In Paper Reading:**

N/A

---

> ### Author Response · Authors · 2019-11-07
> **Thank you so much for going through the paper carefully and providing such a positive feedback**
>
> Dear Reviewer #2,
>
> Thank you so much for going through the paper carefully and providing such a positive feedback. Please see below our response to your comments:
>
> About all-sharing vs non-shared: yes, non-shared gives better results, but the number of parameters is increased dramatically. We tried other strategies of sharing the parameters across layers. For example, we divided the L layers into N groups of size M (L=N*M), and each size-M group shares parameters. Overall, our experimental results show that the smaller the group size M is, the better the performance we get. However, decreasing group size M also dramatically increase the number of overall parameters. We chose the all-shared strategy to maximize our parameter reduction.
> We are glad that you like our SOP objective. We did try other changes to the objectives, such as multiword masking, but papers proposing these ideas were posted before our work, so for simplicity we adopted and cited the previous works.
> Thank you so much for helping us to correct the typo, we will fix them in our next version.

---

### Public Comment · ~Emma_Strubell1 · 2019-09-27
**compare speed to BERT-large not BERT-xlarge, and a comment on dropout**

Thanks for this nice paper, important work! It wasn't clear to me which cross-layer parameters are shared in the final results, it would be great if you could update the paper to make that more clear (perhaps in section 3.1).

In Table 3 I think you should compute speed-up relative to BERT-large, the model that performs well and is actually used, rather than BERT-xlarge, which is not used. This would provide a more practical sense of how your model performs compared to the existing state-of-the-art, rather than a sort of artificially inflated benchmark :)

I also think you should be careful about generalizing your results on removing dropout to large Transformer-based architectures more generally. The cross-layer parameter sharing is a form of regularization that could negate the need for dropout in your architecture/training setup, but this sharing is not typical, and so it remains unclear whether dropout is needed in large Transformers.

---

> ### Author Response · Authors · 2019-10-04
> **Thank you for your comment**
>
> Hi Emma,
>
> Thanks for your comment. I am glad that you like our work.
>
> For the cross-layer parameters sharing technique in the final results, we use all-shared technique as stated in section 3.1: "The default decision for ALBERT is to share all parameters across layers." We will make it more explicit by adding the following sentence: "all our experiments use this default decision unless otherwise specified".
>
> Regarding speedup comparisons: fair enough. The reason we treated BERT-xlarge as a Speedup of 1X in Table 3 is because it's the slowest. To compare to BERT-large instead, one can divide all Speedup numbers by 3.8. We also have direct speed comparison between ALBERT-xxlarge and BERT-large in Section 4.7 and Section 5. However, we take your point that the BERT-large Speedup numbers are more useful for the community and will modify this in the next version.
>
> For the overgeneralization problem in the sentence “dropout can hurt performance in large Transformer-based models”, we will make it clear by adding this sentence: “However, the underlying network structure of ALBERT is a special case of the transformer and further experimentation is needed to see if this phenomenon appears with other transformer-based architectures or not.”

---

### Public Comment · ~Sachin_Mehta1 · 2019-09-30
**Comparison with existing embedding factorization methods**

Enjoyed reading the paper! Interesting results!

I had a question about embedding factorization. It looks like the proposed embedding factorization technique has been proposed in previous related work [r1, r2] for similar computational benefits. It would be great if authors can describe the differences (along with citing these highly relevant papers) of the proposed embedding factorization technique with respect to existing methods. To me, the factorization approach introduced in this paper is a special case of [r1] where the number of clusters is equal to one and has the same size as vocabulary. Also, this factorization looks the same as the one used in [r2] for their base model. See code here:

https://github.com/kimiyoung/transformer-xl/blob/44781ed21dbaec88b280f74d9ae2877f52b492a5/pytorch/mem_transformer.py#L452

[r1] Baevski, Alexei, and Michael Auli. "Adaptive input representations for neural language modeling." arXiv preprint arXiv:1809.10853 (2018).
[r2] Dai, Zihang, et al. "Transformer-xl: Attentive language models beyond a fixed-length context." arXiv preprint arXiv:1901.02860 (2019).
[r3] Grave, Edouard, et al. "Efficient softmax approximation for GPUs." Proceedings of the 34th International Conference on Machine Learning-Volume 70. JMLR. org, 2017.

---

> ### Author Response · Authors · 2019-10-04
> **Thank you for your comment**
>
> Hi Sachin,
> Thank you for your comments, I am glad that you enjoy our paper.
> Although Baevski and Auli [r1], Dai et al. [r2], and Grave et al. [r3] also try to address the large vocabulary problem, they have different settings from the one in our paper for the following reasons:
> 1) Different problem settings:
> [r1][r2][r3] use whole words embeddings as they main setting while we use subword embeddings. Because of this difference, their vocabulary sizes are much larger than the one we have. They have a problem of how to learn good embeddings for those words that rarely occur while this problem is less of an issue for us. Our main problem is the large memory consumption because of the large number of parameters.
> 2) Different main ideas of solutions:
> Because of the different problems we face, we use different solutions. [r1] and [r2] have adaptive embedding size to make it easier for infrequent words to learn embeddings. We use same but smaller embedding size for all our subwords. One evidence that illustrates the differences between these two solutions is that In [r1], they have a comparison with subword embeddings, where they still tie the embedding size and hidden size.
> 3) Different experimental settings:
> [r1][r2][r3] focused on standard language modeling tasks rather than the pretraining/finetuning setting we have.
> Nonetheless, It is helpful to illustrate our idea by comparing embedding factorization methods with adaptive softmax [r2] [r3] and embeddings [r1][r2]. And we agree that these relevant papers are worth citing and including in our paper. Thanks for your useful suggestions!

---

### Public Comment · ~Brandon_Norick1 · 2019-10-11
**Where are the parameters?**

I was investigating the parameter counts across the various ALBERT variants. It seems that 8M parameters appear in the xxlarge model that are unaccounted for.

When calculating the number of parameters, the only differences from a single layer BERT model (with appropriate transformer dimensions) should be the change in embedding dimension and the additional embedding projection. This yields the following formula: $numparams = v*e + e*h + s*h + 2*h + h + h + h*h + h + h*h + h + h*h + h + h*h + h + h + h + h*4*h + h*4 + h * h*4 + h + h + h + h*h + h$, where $v$ is vocab size, $e$ is embedding size, $s$ is maximum sequence length, and $h$ is hidden size. I have used $v=30000$ and $s=512$, then varied the hidden and embedding sizes.

For the three smaller variants (base, large, xlarge) the parameter count matches the formula above, but for xxlarge there are about 8M missing parameters.

$$
\begin{align*}
\textrm{ALBERT}&\textrm{-base} &=& \enspace 12,013,056 \\
\textrm{ALBERT}&\textrm{-large} &=& \enspace 18,145,280 \\
\textrm{ALBERT}&\textrm{-xlarge} &=& \enspace 59,713,536 \\
\textrm{ALBERT}&\textrm{-xxlarge} &=& \enspace 224,638,976
\end{align*}
$$
Can the authors explain this discrepancy?

---

> ### Author Response · Authors · 2019-10-12
> **Thanks for the comment**
>
> Hi Brandon,
>
> Thank you so much for helping us to check the parameter counts! your numbers are correct. The ALBERT-xxlarge discrepancy is due to a typo. We meant to put in 223M as in Table 14  (appendix) instead of 233M. We will update the ALBERT-xxlarge count to be 225M and ALBERT-xlarge count to be 60M.

---

### Public Comment · ~Chen_Yan1 · 2019-10-14
**Official implementation and pre-trained ALBERT?**

I read this nice paper and interested in it. But it seems like no official code of ALBERT on Github. And there’re only some unofficial implementation like https://github.com/brightmart/albert_zh
(which is trained on Chinese vocab)

Where can I find official implementation and pre-trained  models of ALBERT(as in paper)? It may help me to understand the improvement points of ALBERT.

---

> ### Author Response · Authors · 2019-10-14
> **Thank you for your comments**
>
> Hi Chen,
>
> Thanks for your comment. I am glad that you find our work interesting.
>
> We are working on releasing the code as well as the models. As the same time, please let us know if you have any question about where the improvements come from. We will try out best to make it clear to you.

---

> > ### Public Comment · ~Tomotaka_Sasaki1 · 2019-10-17
> > **When will the code and the models be released?**
> >
> > Thanks for this nice paper.
> > I'm interested in ALBERT, so I want to use it.
> >
> > When will the code and the models be released?
> >
> > I'm so looking forward to them.

---

> > > ### Author Response · Authors · 2019-10-17
> > > **Thanks for your comment**
> > >
> > > Hi Tomotaka,
> > >
> > > Thanks for your comment. I am glad that you like our work. We are in the process of releasing the code and models. They are expected to be out late this week or early next week. Please stay tuned!

---

> > > > ### Public Comment · ~Tomotaka_Sasaki1 · 2019-10-21
> > > > **Would you tell me the URL when the code and the models are released?**
> > > >
> > > > Thank you for your comment.
> > > > I'm really looking forward to ALBERT coming soon.
> > > >
> > > > Would you tell me the URL when the code and the models are released?

---

> > > > > ### Author Response · Authors · 2019-10-23
> > > > > **the url to the code and models**
> > > > >
> > > > > Hi Tomotaka,
> > > > >
> > > > > The code and models are out in the public. But I cannot put the URL here due to the anonymous policy. However, if you put your email here, I can send you the link directly. Another way is to search the code online.

---

> > > > > > ### Public Comment · ~Tomotaka_Sasaki1 · 2019-10-25
> > > > > > **my email**
> > > > > >
> > > > > > My email is "ylvwzj@1timl.com".
> > > > > > Would you send me the link?
> > > > > >
> > > > > > I look forward to hearing from you.

---

### Public Comment · ~Shuailiang_Zhang1 · 2019-10-15
**It is DCMN, not DCMI in Table 11.**

The model name DCMI should be DCMN in Table 11.

---

> ### Author Response · Authors · 2019-10-15
> **Thanks for your comment**
>
> Hi Shuailiang,
>
> Thanks for your comment. Will correct this typo in next version.

---

### Public Comment · ~Qingqing_Cao1 · 2019-10-18
**Curious about inference speed over BERT**

Hi, thanks for your excellent work! I'm curious about how much inference speed benefits ALBERT can bring compared to BERT. The paper seems only discussed the training gains; correct me if I'm wrong. Some inference speedup numbers will be good to know.

---

> ### Author Response · Authors · 2019-10-20
> **Thank you for your comment**
>
> Hi Qingqing,
>
> Thanks for your comment and I am glad that you like our work. We haven't throughly tested the inference speed of ALBERT models yet but theoretically speaking, ALBERT won't improve the inference speed.  It is more of an improvement on memory consumption. As stated in the discussion section, we are currently working on improving the inference speed by applying sparse/block attention.

---

### Author Response · Authors · 2019-10-20
**Missing two important citations**

When we mention "full network pre-training (Radford et al., 2018; Devlin et al., 2019) has led to a series of breakthroughs in language representation learning", we miss the following two seminal work:
1) Dai, Andrew M., and Quoc V. Le. "Semi-supervised sequence learning." Advances in neural information processing systems. 2015.
2) Howard, Jeremy, and Sebastian Ruder. "Universal language model fine-tuning for text classification." arXiv preprint arXiv:1801.06146 (2018).

We apologize to the authors of these two papers and will add them to our citation list in our revised version. Thanks for people who bring this to our attention.

---

### Public Comment · ~Erhan_Bilal1 · 2019-10-21
**Projection and batch size**

Thanks for sharing your work. I have a couple of questions:

1. Is the large batch size/LAMB combo essential for the performance of Albert? Bert uses a smaller batch size/Adam. At least in my implementation Albert converges slower when pretraining on Wiki.
2. Are the initial parameters shared between input and output projections?

---

> ### Author Response · Authors · 2019-10-23
> **Thanks for your comment**
>
> Hi Erhan,
>
> Thank you for your comment. Here are the answers to your questions.
>
> 1. Is the large batch size/LAMB combo essential for the performance of Albert? Bert uses a smaller batch size/Adam. At least in my implementation Albert converges slower when pretraining on Wiki.
>
> Not sure how large is the batch size you mentioned, but I tested on the 512/1M step setting from this paper (https://arxiv.org/pdf/1904.00962.pdf) as well and they converge to a roughly similar MLM accuracy.  I also tried to use Adam and the results there are also similar.
>
> Given that none of our models really converge, I am not sure how you define slower convergence. However, albert-base does perform worse than bert-base, so you will see lower MLM accuracy given the same number of iterations.
>
> 2. Are the initial parameters shared between input and output projections?
> Yes. There are shared as BERT does.

---

> > ### Public Comment · ~Erhan_Bilal1 · 2019-10-25
> > **Thanks**
> >
> > Thanks, this explains it. I was referring to MLM accuracy for the same number of steps.

---

### Public Comment · ~Shaojie_Bai1 · 2019-11-06
**Some very minor points on ALBERT and the deep equilibrium models**

Hi,

I'm one of the authors of the deep equilibrium models. Congrats on the great work! I'm extremely excited that there is yet another proof of the power of weight-tied (or in your word, cross-layer sharing) models. I believe further investigation in this direction is a promising avenue for future work :-)

Just a few very very minor points:

- It's actually DEQ (as in Deep EQuilibrium models) instead of DQE :P

- In your work, you claim that "our observations show that our embeddings are oscillating rather than converging", which you imply "shows that the solution space for ALBERT parameters is very different from the one found by DQE [DEQ]". But in fact, we also found that weight-tied Transformers oscillate (see Figure 2, right, and Figure 4, right, of the DEQ paper [1]); so it's not contradictive with your finding. In fact, we empirically found that these weight-shared networks are exactly oscillating around a fixed point (with some cyclic frequency, which is why layer diff seems to "oscillate"). We hypothesize this phenomenon is a result of the limitations of simple fixed-point iterations (FPI), which is what a weight-shared deep network is essentially doing. (The figure in https://aimeemarie88.wordpress.com/fixed-point-iterations/ actually visualizes this phenomenon for FPI, when the operator norm of the Jacobian of the tied transformation approaches 1). What we generally found is that once you target explicitly at solving for this fixed point, you will still be able to find it at the center of such oscillations (which yields DEQ).

But anyway, as I mentioned, these are very minor points. I just feel that this may help clarify a better connection between ALBERT and DEQ. (Plus, of course, in ALBERT you didn't use things like input injection, which DEQ required to formally formulate the implicit-depth model). Again, I'm excited about ALBERT!

[1] https://arxiv.org/pdf/1909.01377.pdf

---

> ### Author Response · Authors · 2019-11-07
> **Thanks for helping us to explain the oscillating phenomenon**
>
> Hi Hi Renjie!
>
> Thanks for commenting on our paper! I enjoy reading your paper and hoped that I could come up with such an elegant solution!
>
> Would address the name error problem and acknowledge that DEQ (you) also found that simple weight-sharing would result in oscillation.

---

### Author Response · Authors · 2019-11-14
**Paper updates**

We want to thank the reviewers again for their suggestions! We have updated the paper with the following changes:
  - Addressing the typo pointed out by Reviewer 2.
  - Addressing Reviewer 3’s concern on the overgeneralization problem of this sentence “dropout can hurt performance in large Transformer-based models”
- Addressing Reviewer 3’s suggestion of comparing embedding factorization with other related methods in the literature.
- Addressing Reviewer 3’s suggestion clarifying the additional data usage.
- Addressing all other public comments.

We also move the section of comparing BERT-xxlarge to BERT-large with the amount of training time to appendix so that we can fit the 10 page limit.

---

### Decision · Program_Chairs · 2019-12-19

**Decision:**

Accept (Spotlight)

**Comment:**

This paper proposes three modifications of BERT type models two of which is concerned with parameter sharing and one with a new auxiliary loss. New SOTA on downstream tasks are demonstrated.

All reviewers liked the paper and so did a lot of comments.

Acceptance is recommended.